# Quality of Work Life and Organizational Performance: Workers’ Feelings of Contributing, or Not, to the Organization’s Productivity

**DOI:** 10.3390/ijerph16203803

**Published:** 2019-10-10

**Authors:** João Leitão, Dina Pereira, Ângela Gonçalves

**Affiliations:** 1Universidade da Beira Interior, 6201-001 Covilhã, Portugal; dina@ubi.pt (D.P.); angela.goncalves@ubi.pt (Â.G.); 2NECE–Research Center in Business Sciences, 6200-209 Covilhã, Portugal; 3CEG-IST–Centro de Estudos de Gestão do Instituto Superior Técnico, 1049-001 Lisboa, Portugal; 4ICS–Instituto de Ciências Sociais, 1600-189 Lisboa, Portugal

**Keywords:** organizational performance, productivity, quality of work life

## Abstract

This is a pioneering study on the relationship between quality of work life and the employee’s perception of their contribution to organizational performance. It unveils the importance of subjective and behavioral components of quality of work life and their influence on the formation of the collaborator’s individual desire to contribute to strengthening the organization’s productivity. The results obtained indicate that for workers: feeling their supervisors’ support through listening to their concerns and by sensing they take them on board; being integrated in a good work environment; and feeling respected both as professionals and as people; positively influence their feeling of contributing to organizational performance. The results are particularly relevant given the increased weight of services in the labor market, together with intensified automation and digitalization of collaborators’ functions. The findings also contribute to the ongoing debate about the need for more work on the subjective and behavioral components of so-called smart and learning organizations, rather than focusing exclusively on remuneration as the factor stimulating organizational productivity based on the collaborator’s contribution.

## 1. Introduction

Employee workplace performance is related to a set of factors affecting workers’ health, habits and environment, employees’ well-being and quality of work life (QWL). QWL is associated with job satisfaction, motivation, productivity, health, job security, safety and well-being, embracing four main axes: a safe work environment; occupational health care; appropriate working time; and an appropriate salary [1]. As originally stated in [2], the concept embraces the effects of the workplace on job satisfaction, satisfaction in non-work life domains, and satisfaction with overall life, personal happiness and subjective well-being. Moreover, improving employees’ QWL will positively affect the organization’s productivity, while augmented productivity will strengthen QWL [3].

In the literature of reference, there is an ongoing and fruitful discussion about the components of QWL [3] and its different associations with metrics of non-economic performance, namely satisfaction and fulfillment of physical conditions considered basic to ensure functionality, health and safety in the workplace [1].

The most sensitive components of the QWL, still unexplored, are intrinsically related to the socio-emotional and psychological needs of employees, which require the application of more behavioral lenses, in order to unveil the components that can most influence job satisfaction and motivation, but also productivity [4,5].

In the context of health organizations, the relationship between QWL and productivity was already investigated, suggesting the design of adequate strategies to reinforce the productivity in hospitals [6]. However, little is known about the different ways in which the behavioral and subjective components of the QWL can influence the employee’s feeling of contribution to the productivity of the organization that they integrate.

As stated before, there is still room to advance knowledge about the effects associated with subjective components of assessment of satisfaction with QWL on organizational performance, considering a response variable of particularly critical importance in the context of reducing investment in resources and simultaneous pressure to maximize results, i.e., productivity [7]. Therefore, it is particularly opportune to investigate the non-economic (that is, subjective or behavioral) motivations that lead to collaborators’ willingness to contribute to strengthening their organization’s productivity.

Following the Organisation for Economic Co-operation and Development (OECD)’s view of productivity indicators, there are plenty of productivity differences across organizations that require further studies to open up the organizational ‘black box’, concerning internal productivity determinants [8]. In fact, there is a need to advance knowledge about the individual determinants of organizational productivity. An example of this challenging task is the recent project launched by the Global Forum on Productivity (GFP), entitled: ‘The Human Side of Productivity’; considering a multidimensional approach applied to organizations, considering key people, such as workers, managers and owners [9].

Recently, in the context of public higher education, the role played by quality of life in determining satisfaction of internal stakeholders, such as students and collaborators (e.g., administrative staff, teachers and researchers), was also assessed. This opens up a research avenue concerning the lack of knowledge about the role played by the specificities of different organizational cultures in this type of institution, in influencing perception of academic quality of life by both internal and external stakeholders [10].

In this sense, there is still an open debate about the need for further understanding of the importance of organizational culture, using crossed perspectives on organizational and individual health, to be able to provide strategic lines for new organizational policies. These should be increasingly funded on a particular set of values and beliefs determining an organization’s behavioral objectives, aligned with the desired self-efficacy in terms of employees’ management and motivation [11]. 

Following this debate, the current study is particularly relevant, from the view that there is still limited knowledge about the necessary conditions to promote the subjective or behavioral components of satisfaction with QWL, focusing on each collaborator’s contribution to fostering the organization’s productivity. For example, a myth revisited here, through lack of thorough existing knowledge, is that productivity depends mainly on the remuneration attributed to performing certain functions. As yet unexplored subjective or behavioral factors, such as the collaborator feeling appreciated by the supervisor, the availability of jobs not subject to routines and where innovation is possible, promotion of continuous learning environments, the feeling of protection promoted by the supervisor, the feeling of having a really important and useful job, the possibility of the job allowing the development of new skills and reinforcing the conditions for personal and professional growth, are given special attention in this study. A data survey, which is pioneering in European terms, is followed by statistical and econometric treatment to shed new light on a little-explored relationship. i.e., the relationship between QWL and organizational performance, using a subjective measure of assessment of satisfaction expressed through collaborators’ feeling of contributing to organizations’ productivity.

Despite the limitations associated with the use of this dependent variable with subjective nature, its use seems to be justified, on the one hand, given the lack of studies using the behavioral lens to study the relationship between QWL and organizational performance. On the other hand, as it is not the objective of the present study to compare the relationships and the associated significance, using objective measures versus subjective measures, for the purposes of representing the dependent variable: organizational performance.

In turn, the current study aims to reveal employees’ satisfaction with the opportunities and conditions provided by their employer in six European countries, by looking after their QWL and their interests in pursuing a healthier, more satisfactory and happier lifestyle, as well as how the workplace can provide opportunities for them to improve productivity.

This study contributes to the literature on QWL and organizational performance in two ways, firstly, by identifying the determinant factors that can have a significant influence on employees’ understanding of their contribution to organizational performance, represented here by an alternative measure regarding the contribution to organizations’ productivity. Secondly, it provides new insights into complete fulfillment of the functions of human capital managers, revealing the importance of subjective and behavioral components of QWL that can help to design desirable collaborator behavior more likely to strengthen productivity in the organizational context.

The research partners involved in the survey design and administration developed an innovative tool to gather information for assessment of QWL. Afterwards, the survey was administered to 514 employees of local companies and public organizations in six European countries. Some highlights from the preliminary results obtained from the survey’s administration can be illustrated as starting points for the current study. Namely, 80% of respondents said they feel physically safe at work and more than 77% are satisfied with the fact that their workplaces are safe and sanitary. Almost 82% of respondents feel that their organization matches their skills with the needs of their jobs and 76% are satisfied with their workplaces’ maintenance/cleaning conditions. A substantial group (80%) of employees feel they are contributing to the organization’s productivity, and the great majority (83%) of employees revealed that having an important job is extremely important to be productive.

The first impression is that the collaborators seem to be aware of the importance of standard human capital management procedures and conditions oriented to the reinforcement of organizational performance. Nevertheless, it is worth noting that there is a need to address an organizational ‘black box’, an aim of the current study, that is, the set of subjective and behavioral components to promote QWL that can directly influence employees’ feeling of contribution to organizational performance, especially concerning productivity.

The remainder of the paper is structured as follows. After a literature review leading to formulation of the research hypotheses, the research methodology is presented. Next, the results are discussed, followed by the conclusions, limitations and implications.

## 2. Literature Review and Research Hypotheses

### 2.1. Revealing the Relationship between Organizational Performance and QWL

There is no simple or universally recognized definition of what performance is at the level of an individual organization. Organizational performance is multidimensional, connected to its goals and objectives, and may be defined as an organization’s ability to use its resources efficiently, and to produce outputs that are consistent with its objectives and relevant for its users [12]. Analyzing organizational performance is a crucial step in the organizational assessment process [13]. In doing so, in the literature of reference, three main domains of organizational performance have been reported, namely: financial performance; operational performance; and organizational effectiveness [14]. Concerning the conceptualization of organizational performance, four main elements should be taken into consideration: effectiveness; efficiency; relevance; and financial viability [13].

People are the organization’s most important asset [15], and so the way an organization manages people’s impacts has a major influence on organizational performance [16].

Performance management is a continuous process of identifying, measuring and developing the performance of individuals and teams and aligning performance with the organization’s strategic goals [17,18]. The previous arguments are examples of cornerstone visions regarding the need to advance the knowledge available on subjective and behavioral components affecting the relationship between organizational performance and QWL.

Nevertheless, various performance management systems are found in the literature and these systems have some advantages, such as: increased motivation to perform; increased self-esteem; managers gain insights into subordinates; organizational goals are made clear; employee misconduct is minimized; organizational change is facilitated; motivation and commitment to stay in the organization are increased; and employee engagement is enhanced [19]. In fact, performance management systems are the source of information when making decisions about rewards and the allocation of resources, succession planning and staffing strategies [20].

Each employee’s emotional intelligence has an effect on behavior which ultimately affects achievements and performance in the workplace [21]. The satisfaction of employees’ needs through organizational development is at the core of the QWL movement [22]. Enhancing QWL will result in improved productivity, and in turn, gains in productivity will strengthen QWL [3].

Improving QWL and performance is of extreme importance, as productivity and innovation are part of the political agenda of European Union countries. With fewer people in the workforce due to an aging population there is a need to enhance labor productivity [23]. The quality of work life is covered in the guidelines for the employment policies of member states [24].

Previous applied empirical work [25] pointed out the existence of a positive and significant relationship between QWL and organizational performance, as well as a positive and significant association between QWL and employees’ job satisfaction.

Another study [26] found that employee commitment partially mediates the relationship between QWL and organizational performance; and also unveiled that work environment significantly affects employee commitment and thus organizational performance. It was also advocated that improving the QWL of an organization could achieve a heightened job satisfaction, commitment and also improved performance [27]. In order to achieve a higher employee commitment and consequently a better organizational performance, it is suggested for managers to pay attention to the different dimensions of QWL [26].

In contrasting terms, previous scholars [28] reported a negative but non-significant relationship between QWL and organizational performance, although it was also found a positive relationship between employee’s job satisfaction and organizational performance. This type of mixed evidences raises the interest for advancing knowledge about still unexplored subjective and behavioral components of the QWL and their influence on organizational performance.

### 2.2. Exploring Subjective and Behavioral Components of QWL

Quality of life is an elusive concept regarding the assessment of societal or community well-being from specific evaluation of individual or group cases [29]. The literature has associated a high quality of life with higher levels of productivity at the workplace. Therefore, increasing attention has been paid to the role played by occupational stress, including job demands, job control, job insecurity, organizational justice, intra-group conflict, job strain, effort-reward imbalance, employment level and shift work. In turn, this has been correlated with factors that negatively affect quality of life, namely insomnia, which results in impaired work performance and leads to significant productivity losses for organizations [30].

Quality of life is modulated by a wide range of factors, among them psychosocial parameters, health conditions and well-being in the workplace, as well as the adequacy of working resources and infrastructures provided. Policies and regulations created based on employees’ individualized considerations have suggested significant productivity improvement due to subjective components, such as trust, commitment, satisfaction and control. Nevertheless, the research opportunity remains to deepen knowledge about the role played by both subjective and behavioral components of QWL.

For instance, social support, reflecting individuals’ integration into a social group, has been reported as an important indicator of quality of life in occupational performance [31]. Infrastructures also have an important role in providing well-being in the workplace and therefore modulating the quality of life. It has been suggested that providing green lawns in urban areas enhances quality of life in the workplace, maximizing employees’ social interaction, physical activity and connection with nature [32]. Shiftwork has been reported as worsening the quality of life [33].

Cooperative decision making, adequate recognition and supportive supervisors are considered fundamental to QWL [34], with appropriate job performance feedback and favorable relations with supervisors being said to have a direct impact on QWL [35]. Another study [36] goes further and reveals that supervisory behavior is the most important component of QWL, contributing to the variance in the employee’s role efficacy by as much as 21%.

Considering the previous statements in the literature, the following research hypothesis is derived:

**Hypothesis** **1** **(H1).**
*Workers who feel that they are supported and appreciated by their supervisors are more likely to feel that they contribute to the organization’s productivity.*


QWL is considered a multi-dimensional construct with no clearly accepted definition of the term. This subjective definition means accurate measurement of its parameters is complex. QWL differs from job satisfaction [2], as job satisfaction is considered one of the outcomes of QWL. In turn, QWL is mainly associated with job satisfaction, motivation, productivity, health, job security, safety and well-being [37].

Following [1], QWL involves four major parts: a safe work environment; occupational health care; appropriate working time; and fitting salary. According to [2], QWL involves the effect of the workplace on satisfaction with the job, satisfaction in non-work life domains, and satisfaction with overall life, personal happiness and subjective well-being.

The factors relevant to employees’ QWL include the social environment within the organization, the relationship between life on and off the job, the specific tasks they perform and the work environment [38].

Providing safe and healthy working conditions aims to ensure the employee’s good health, thus, taking measures to improve QWL is expected to increase employee’s motivation ultimately leading to the enhancement of performance and productivity [38].

Accordingly, a work environment that is able to fulfill the employee’s personal needs will lead to an excellent QWL [39].

Thus, the following research hypothesis is considered:

**Hypothesis** **2** **(H2).**
*Workers who feel that they are integrated in a good working environment are more likely than others to feel that they contribute to the organization’s productivity.*


Researchers have proposed differentiated models concerning QWL. For example, in [39] a model is proposed in which the needs of psychological growth were connected to QWL. The same authors recognized several needs: skill variety; task identity; task significance; autonomy; and feedback.

In [2], a model is originally proposed founded on five critical key-factors concerning the satisfaction of workers’ needs, namely: (i) work environment; (ii) job requirements; (iii) supervisory behavior; (iv) ancillary programs; and (v) organizational commitment.

The second vision is highly valued in organizations committed to playing a responsible role in society, since QWL benefits the employee’s pride, social commitment, satisfaction and the organization’s contribution to society [11,40]; and can also be positively influenced by organizational support, for instance by relieving fatigue and enhancing self-efficacy [41].

QWL has been considered as the condition experienced by the individual in terms of the dynamic pursuit of their hierarchically organized goals within work domains, whilst reducing the gap separating the individual from these goals can have a positive impact on the individual’s general quality of life, organizational performance, and consequently on the overall functioning of society [42].

Furthermore, QWL is a phenomenon that can originate a change in terms of organizational culture, since the former corresponds to employees’ interpretation of all the conditions in a workplace and their perception of those conditions [43].

In a related vein, QWL can be approached as an indicator of the overall quality of the human experience at work [44]. The same author advocates that it creates a favorable workplace, which enhances employee well-being and satisfaction.

Employees that feel they are treated with respect by people they work with, and employees who feel proud of their job, increase their feeling of belonging to the company, thus feeling that they are an asset to the organization [45]. Studies [46,47] found that feeling respected is a predictor of QWL, together with self-esteem, variety in daily routine, challenging job, autonomy, safety, rewards and good future opportunities; and as already mentioned an improved QWL is expected to lead to a higher productivity [48].

Considering the previous vision, the QWL construct can be completed by incorporating subjective measures related with employee satisfaction, motivation, involvement and commitment with respect to their lives at work [49]. In the same vein, QWL corresponds to the degree to which individuals are able to satisfy their important personal needs while employed by the firm. This gives rise to the following research hypothesis:

**Hypothesis** **3** **(H3).**
*Workers who are respected as professionals are more likely than others to feel that they contribute to the organization’s productivity.*


Employees can experience a better QWL if they have a positive perception of the degree of responsibility of the organization they belong to [50]. A related study about perceived QWL in Croatia found that employees positively value non-competitive, co-operative work environments for improved quality of life [51]. In addition, factors like job security, human relations and work-life balance influence QWL positively [52]. The analysis of the first European Quality of Life Survey found also that positive aspects of work (good rewards, job security, favorable career prospects and interesting work) have a greater impact on life satisfaction and particularly job satisfaction [53]. In turn, it should be noted that a poor work-life balance lowers employees’ quality of life [53].

Work-life balance has been positioned in the reference literature as a key component of QWL [38,54,55,56,57,58], but it deserves to be noted that the employee’s level of emotional intelligence could influence his/her work-life balance [59].

It should be noted also that in a previous empirical study [60] no significant association, neither positive nor negative, between work-life balance and productivity was detected.

Nevertheless, Work-life balance plays an important role in overall life satisfaction and influences experiences in work life by increasing job satisfaction and organizational commitment [61]. A high level of engagement in work life is likely to produce a positive effect in work-life balance, which can be further enhanced by goal attainment in work life [62]. Accordingly, the following research hypothesis is derived:

**Hypothesis** **4** **(H4).**
*Workers who have the possibility to enjoy the adoption of work-life balance practices in their organizations, are more likely than others to feel that they contribute to the organization’s productivity.*


QWL involves acquiring, training, developing, motivating and appraising employees in order to obtain their best performance, in accordance with the organization’s objectives [28]. QWL is the foundation of employee well-being and leads to better performance [26].

Skills, occupational improvement and opportunity for training are considered sub components of QWL [45,63,64]. The development of skills and abilities can improve job satisfaction and overall QWL, and for its turn QWL can influence the employee’s performance [65,66]. Thus, employees expect to develop their skills and get promoted, ensuring a better performance for the organization [67]. In turn, training is an activity aimed at enhancing performance, by ensuring the opportunities for development of skills and encouragement given by the management team [38].

As previously revealed through the empirical evidence obtained in [68], both QWL and motivation influence employees’ performance positively. High levels of QWL lead to job satisfaction, which ultimately results in effective and efficient performance [49]. Considering the previous statements and empirical evidence, the following hypothesis is derived:

**Hypothesis** **5** **(H5).**
*Workers who feel that their organizations invest in their careers, for example through continuous learning, the development of new skills or supporting professional growth, are more likely to feel that they contribute more than others to the organizations’ productivity.*


## 3. Empirical Approach

### 3.1. Methodology and Data Characterization

The research methodology was developed using different questionnaires, which were designed taking into consideration a set of eleven selected international benchmarks, namely: (i) Health and well-being at work: a survey of employees, 2014, UK, Department for Work and Pensions; (ii) ACT Online Employee Health and Wellbeing Survey 2016, Australian Capital Territory Government; (iii) British Heart Foundation 2012, Employee survey; (iv) British Heart Foundation 2017, Staff health and wellbeing template survey; (v) Rand Europe (2015), Health, wellbeing and productivity in the workplace—Britain’s Healthiest Organization summary report; (vi) South Australia Health, Government of South Australia Staff needs assessment, Staff health and wellbeing survey; (vii) Southern Cross Health Society and BusinessNZ, Wellness in the Workplace Survey 2017; (viii) State Government Victoria, Workplace Health & Wellbeing needs survey; (ix) East Midlands Public Health Observatory, Workplace Health Needs Assessment for Employers, February 2012; (x) Tool for Observing Worksite Environments (TOWE). U.S. Department of Health & Human Services; and (xi) Measure of QWL, as originally proposed in [2].

The survey was conducted from April to July 2018. Twelve partners from Italy, Bulgaria, Cyprus, Portugal, Greece and Spain participated in data collection, by interviewing employees. The sample covers 15 private companies and five public entities or large firms per partner, involving two employees per organization and totaling 514 questionnaires. It was not intended to interview company owners or general managers to avoid bias in the responses.

A convenience sample procedure based on random selection was used. In each organization, a contact person was identified to ensure completion of the questionnaire, which was afterwards validated by the research team. The questionnaires were applied by personal interviews to ensure a maximum response rate.

The partners followed the following instructions in selecting interviewees: 15 companies among micro, small and medium-sized firms (10% of interviewees for each category—EU definition of SME), plus five among large firms and public entities.

The main aim of the study is to assess the influence of workers’ QWL on the perception of their contribution to organizational performance. The degree of novelty here lies in the innovative assessment of both subjective and behavioral components of workers’ QWL, embracing different types of organizations (e.g., public or private) with distinct dimensions and economic activities. A total of 514 questionnaires were collected involving organizations from the six European countries engaged in the data collection process.

The questionnaire includes two sections: (1) QWL (needs, work environment, work requisites, supervisor behavior, auxiliary programs inside the organization, organizational pressure, and organizational performance and commitment); and (2) sample characterization (gender, age, marital status, position in the organization, level of qualifications, organization’s sector of activity, size and age of the organization, type of employee contract and employee qualifications). In the first section, Likert scales (e.g., ranging from 1 to 7) were used to assess the level of agreement with a set of sentences in each sub-section, scales that had been transformed into binary considering the variables under analysis, namely the Feeling of contributing to productivity, Supervisors’ support, Good work environment, Professional respect and Work-life balance. In the second section, levels of answer were used. Below, the sample is characterized and a set of results for the whole sample is presented.

### 3.2. Sample Characterization

#### Sample and Descriptive Statistics

Concerning respondents’ gender, 48% were women and 52% men. Relative to age, 9% were aged between 20 and 25, 34% between 26 and 35, 37% between 36 and 45, 14% between 46 and 55 and only 7% were older than 55. 35% were single, 59% married and almost 7% are in another non-defined situation. In terms of organizational role, 18% said they occupied a managerial role inside the organization, 67% a qualified role and 16% a non-qualified position. Regarding education, 51% have a college degree and 22% a post-graduate degree, 19% completed secondary education, 7% completed 9 years at school and only 1% completed 4 years. Concerning the sector of activity of the respondents’ organizations, almost 2% belong to the primary sector, 14% to the secondary, 77% to the tertiary and 7% to public organizations. The majority of respondents work in small and medium sized firms, 26% in companies with one to nine employees, 39% in firms with 10 to 49, 15% in companies with 50 to 249, 14% in companies with 250 to 1000 and 6% in companies with over 1000 employees. Concerning the organizations’ age, 16% are between 1 and 6 years old, 34% between 7 and 15 years, 25% between 16 and 29, almost 17% between 30 and 49 years and almost 8% have been in existence for more than 50 years. Concerning respondents’ contract type, 68% said they have a permanent contract, 11% a contract for a stipulated period, almost 9% were temporary, 5% were freelancers and 9% reported another sort of contract. Lastly, respondents were asked about their qualification inside the firm, with almost 7% saying they were senior managers, 10% intermediary managers, almost 17% staff in charge, 21% highly qualified employees, approximately 25% qualified, 6% semi-qualified and 8% non-qualified. In addition, 3% said they were apprentices and 1% said they did not know.

In descriptive terms, for the employees, it is observed that the items in which they feel more in agreement in their workplaces are professional respect as workers and people (70%), followed by the existence of a good work environment (65%), as seen in Table 1 presented below. For 62% of respondents having the supervisors’ support is essential. Approximately 37% denote the importance of having a work-life balance and 57% show that the organizations’ support for skills development is essential. Approximately 80% of the workers feel they really contribute to the organization’s productivity. Looking at the correlations matrix we can observe that the items most associated with the workers’ sense of contribution to the organizations’ productivity are professional respect, having a good work environment, and lastly supervisors’ support.

The variables presented above were subsequently used in estimation processes, considering two distinct models: (1) an Ordinary Least Squares (OLS) model; and (2) a Multinomial Logit model; in order to reveal the set of subjective and behavioral components of QWL that influence the workers’ perception of contribution to productivity. The main reasons for using the two models are as follows: (i) estimation of the OLS model is justified by the dataset analyzed following normal distribution, considering a dependent variable represented in binary terms, which can determine the probability of the influence of a hypothetical set of independent variables arising from the literature review presented above; the dependent variable takes the value of 1, when the employee states they feel they contribute to productivity; and 0, otherwise; and (ii) estimation of the multinomial model can test a representation at level of the same dependent variable, which lets us, first, contrast the empirical evidence with Model 1, and secondly, determine the variability of the probability of influence of the same hypothetical set of independent variables, through comparison of the results between a baseline corresponding to: ‘not contributing to productivity’ (level 1); ‘contributing to productivity to some extent’ (level 2); and ‘totally contributing to productivity’ (level 3).

To do so, the log-odds for these two categories relative to the baseline are computed, and then the log-odds are considered as a linear function of the predictors. Several control variables were used, namely: gender; age; marital status; employee’s role; employee’s education; organization’s sector; organization’s size; organization’s age; and employee’s position in the organization. The operational model of analysis is as follows (Figure 1):

Table 2 below presents more details and description of the set of variables.

## 4. Results and Discussion

Regarding the results of the OLS regression for the sample considered (see correspondent column of Model 1, in Table 3), which used as dependent variable the feeling of contribution to productivity, with the value of *1* when the worker declares they feel they contribute to productivity and 0 otherwise, the LR Chi^2^ of 14.38 with a *p*-Value of 0.0000 indicates that the model as a whole is statistically significant.

As observed in Table 3 below, three statistically significant variables influence workers’ sense of contribution to productivity, namely: (i) professional respect; (ii) having a good work environment; and (iii) feeling supervisors’ support. Interestingly, work-life balance and the organization’s skills development support do not have any significant influence on the feeling of contribution to the organizations’ productivity.

Moreover, from the control variables tested in the first model, it should be noted that employees’ college education level has a significant and positive effect on their feeling of contribution to productivity.

In Model 2, the likelihood ratio quotient of 22.06 with a *p*-Value of 0.0002 signals that the model as a whole is statistically significant. Here, a set of predictors related to collaborators’ sense of contribution to productivity (computing a categorical variable with three levels: 1, not contributing to productivity; 2, contributing to productivity to some extent; and 3, totally contributing to productivity; are considered in the empirical application.

Regarding the sense of contributing to some extent to organizations’ productivity, only work-life balance denotes a significant, although negative, influence. Moreover, the older the workers are the more likely they are to feel somehow productive to their organizations. Concerning level 3, representing the feeling of totally contributing to the organization’s productivity, workers feeling respected by their companies, sensing that their organizations make them feel confident and value their contribution affects in a positive and significant way the high level of feeling they contribute to firms’ productivity. Workers who feel they are highly productive are also older and those occupying managerial roles and direction positions in their organizations

Contrasting the two estimation processes, we conclude that the OLS model reveals most predictors explaining workers’ feeling of contribution to productivity, by detecting positive and significant influences of 3 out of 6 subjective and behavioral components of QWL. Going deeper, it is important to crosscheck what predicts the collaborator’s feeling of lack of contribution to productivity, in order to improve the management capacity of human capital, following a behavioral approach.

Bearing in mind the set of research hypotheses under examination, new insights arise concerning the subjective and behavioral components of QWL influencing employees’ feeling of contribution to productivity.

Thus, model 1 gives support to H1a, as workers who feel they are supported and appreciated by their supervisors feel they contribute more to the organizations’ productivity than others. These findings are in line with prior findings of [30], stressing the importance of workers being supported and appreciated for increased productivity.

Model 1 supports H2, as we detect a significant and positive influence of good workplace environments, by being safe and sanitary, on workers’ feeling of productivity. Such results are aligned with prior studies which detected a positive association between job security, safety and well-being at the workplace and job productivity, satisfaction and motivation [37], and the existence of a safe work environment and its positive impact on productivity [1]. These results are aligned with prior literature, which found that by being involved in a socially supportive group inside the workplace, employees are more likely to contribute to organizational performance [31]. In the same line of reasoning, a study referred to previously, applied to the Croatian context [51], identified an important impact of co-operative working environments on QWL.

We found support for H3, as workers who feel respected as professionals (in Models 1 and 2) contribute more to organizations’ productivity than others. In Model 1, our empirical findings reveal a positive and significant influence of workers being professionally respected on the sense of feeling productive. Regarding the findings of Model 2, this influence is also important but only for the group of workers who feel they contribute greatly to the organization’s productivity. This corroborates the rationale of the model proposal found in [39], which outlined that the needs for psychological growth covering the different frameworks associated with professional valorization and respect (namely, skill variety, task identity and significance, autonomy and feedback) are connected with QWL and thus performance. Moreover, our results ratify the concluding remarks of previous scholars [11,40], who defended that employees’ sense of pride and commitment, in relation to being valued as professionals, increases their contribution. These visions are also in agreement with previous empirical findings denoting a positive effect of the worker being considered and taken into consideration in the organizations’ goals on performance [42].

Concerning H4, which states that workers who have the possibility to enjoy the adoption of work-life balance practices in their organizations, feel they contribute more to the organizations’ productivity than others, no significant evidence is found in Model 1. Moreover, in Model 2 we detect a significant, although negative, effect of employees’ feeling that the organization has a work-life balance vision on the feeling of contributing to productivity and so this hypothesis is rejected. This can be justified by the lack of work-life balance practices on the part of supervisors and the organization itself, as well as possible development of a negative emotion concerning the work-life balance allowance, which in certain organizational contexts could be interpreted as a mode of diminishing the potential leadership responsibilities given to target-workers.

The results are contrasting, but do not reject the previous findings in [52], which argued for a positive association between work-life balance and quality of work life, thus spurring productivity. In a similar vein, achieving a balance between private and professional life is expected to be positively associated with organizational commitment and, thus, with productivity at work [61]. In fact, the empirical findings obtained here not only do not contradict the previously identified positive association between work-life balance and QWL, but also shed some light on ‘invisible ceiling’ issues related with the gender leadership issue and supervisors’ behavior within the organizational context, which need to be further explored in future research concerned with organizational productivity based on the individual behavior (of supervisors and workers) and subjective well-being influenced in the scope of the organizational context’s boundaries.

We found no support for H5, stating that workers who feel their organizations invest in their careers and skills development, for example through continuous learning, the development of new skills or supporting professional growth, contribute more to organizations’ productivity than others. Interestingly, our findings do not seem to be related with prior work, for example, in [39], which pointed out an association between professional valorization (skill variety), QWL and performance, as well in [28], where positive argumentation was given to reinforcing investment in employees’ training, to be able to achieve better performance levels in the future. This contrasting result could be justified by the productivity measure used, being a subjective measure, concerning the perception of being productive. These results also contrast with prior literature defending a positive association between organizational investment in workers’ management and organizational performance [16], as well as paying attention to employee management systems, aligning the goals of the organization with career decisions, rewards, structured growth and thus impacting positively on workers and organizations’ performance.

## 5. Conclusions

This study analyses, in an innovative way, the influence of subjective and behavioral components of QWL on organizational performance, measured through collaborators’ feeling of contribution to the organization’s productivity. The empirical findings show the importance of factors related with workers having their supervisors’ support, integration in a good work environment and feeling respected both as professionals and as people.

One of the research challenges addressed here, in a pioneering way, is the use of a subjective measure of collaborators’ commitment to organizational productivity, attempting to provide new implications for organizational management, taking into account components that were hitherto unexplored empirically, various subjective and behavioral components that require greater knowledge to address, in an alternative way, improved organizational performance and behavioral drivers of productivity, rather than relying exclusively on increasing collaborators’ remuneration.

Adopting a more behavioral line of organizational management, and integrating the emerging literature on the QWL construct originally proposed in [7], this analysis contributes to the literature on QWL and organizational performance, bringing two axes of reasoning founded on new empirical evidence, namely: (1) identifying factors that can influence organizational performance, represented here by an alternative measure referring to the collaborator’s feeling of contributing to the organization’s productivity; and (2) proposing a new agenda for human capital managers, focusing on the importance of subjective and behavioral components of QWL, which can help to strengthen productivity in the organizational context, following a behavioral approach both at the company and individual level.

Regarding implications, the evidence obtained signals that human capital managers committed to reinforcing organizational productivity through changing the behavior of collaborators and the organization itself should seek to fulfill a new strategic action agenda with the following priorities: (1) fostering an organizational culture that values behavioral practices of supervisor respect for the collaborator (i.e., hierarchical subordinates) in the organizational context; (2) promoting positive emotions and feelings in collaborators that they are appreciated in the workplace; (3) ensuring that supervisors protect collaborators from hazardous conditions, to reduce feelings of uncertainty and risk; and (4) giving importance to the duties and tasks performed by collaborators.

Surprisingly, this study does not present additional evidence to the established view pointing towards the importance of having a work-life balance and companies’ support for workers’ skills development in the contribution to workers’ productivity. This may be justified, on the one hand, by the content of the research question included in the original survey used in the current study that allows us to point out a hypothetically negative feeling concerning the leadership responsibilities given to target workers, without valuing in a proper way the required work-life balance. Nevertheless, there is still great room for improvement as regards promoting the subjective conditions tending to strengthen behaviors oriented towards stimulating organizational productivity, especially, addressing gender issues, balanced management of the trade-offs between personal and professional life; and leadership responsibilities, per gender role.

The main limitations of the analysis concern the impossibility of carrying out a study with a time dimension, which could determine hypothetical relationships of causality (or precedence) between subjective and behavioral components and organizational performance. Another limitation is in relation to the response variable representing organizational productivity being based on a subjective measure of the collaborator’s perception of individual contribution to organizational productivity. Nevertheless, considering the difficulty in obtaining data of a subjective nature and the aims of this study, it seems acceptable to consider this alternative measure of the organization’s non-economic performance, which requires future exploration through additional research.

In a related vein, this opens an avenue for tracing further research endeavors, expanding both the number of objective and subjective metrics, in order to gauge the hypothetical differences in the relationships established between QWL’s components and organizational performance, “measured” in objective or subjective terms. This would imply the design of a new questionnaire targeted to assess the feelings of the leaders regarding the performance of workers, and, afterwards, it will be possible to produce a contrasting analysis.

For the future, more thorough study of the relationship between QWL and organizational productivity is suggested, by making a comparative analysis involving different profiles of organizational culture considering other contexts of organizational location, for example, in America, Asia, Europe, Africa and Australasia. In this line of analysis, it would also be interesting to pursue this topic considering different organizational and corporate governance contexts, for example, multinationals, family control, female management, management with ethnic diversity and management with values. Another avenue of future research would be the possibility, in the organizational context, of using new forms of organizational design and management able to change behavior in a subjective, inclusive and participatory way. It is necessary, therefore, to explore how design thinking, organizational gamification and co-creation can mobilize the collaborator to contribute effectively to improved organizational performance.

## Figures and Tables

**Figure 1 ijerph-16-03803-f001:**
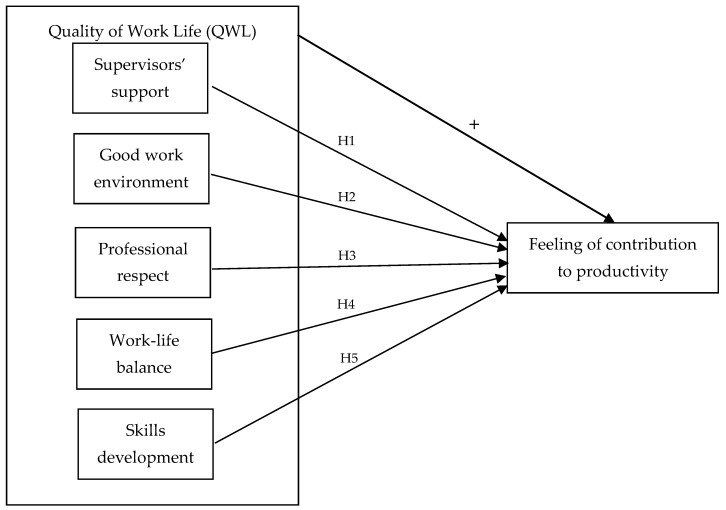
QWL and Feeling of Contribution to Productivity: Operational model of analysis (Source: Own elaboration).

**Table 1 ijerph-16-03803-t001:** Descriptive Statistics and Correlation Matrix.

Variables	M	SD	Skewness	Kurtosis	1	2	3	4	5	6	7	8	9	10	11	12	13
1. Feeling of contribution to productivity	0.8015564	0.3992165	−1.517	0.301	1.0000												
2. Supervisors’ support	0.618677	0.4861848	−0.49	−1.767	0.2722 ***	1.0000											
3. Good work environment	0.6536965	0.4762548	−0.648	−1.586	0.2735 ***	0.3715 ***	1.0000										
4. Professional respect	0.6964981	0.460218	−0.857	−1.27	0.2869 ***	0.3878 ***	0.3911 ***	1.0000									
5. Work-life balance	0.3735409	0.4842151	0.524	−1.732	0.1724 ***	0.2999 ***	0.2662 ***	0.3085 ***	1.0000								
6. Skills’ development	0.5680934	0.4958241	−0.276	−1.931	0.2161 ***	0.2777 ***	0.3064 ***	0.3299 ***	0.3079 ***	1.0000							
7. Female	1.515564	0.5002446	−0.062	−2.004	−0.0333	−0.0477	−0.0346	−0.0641	0.0001	0.0272	1.0000						
8. Age	2.745136	1.01798	0.371	−0.227	0.0624	−0.0038	0.0387	0.0759*	−0.0042	0.0402	0.0824 *	1.0000					
9. Married	0.5603113	0.4968328	−0.244	−1.948	0.0310	−0.0095	0.0143	−0.0221	−0.0371	−0.0048	0.0197	0.4640 ***	1.0000				
10. Manager role	0.1770428	0.3820768	1.697	0.884	0.0774 *	0.1438 ***	0.1341 **	0.1177 ***	0.0738 *	0.1060 *	0.1028 *	0.1012 **	0.1131 *	1.0000			
11. College education	0.7256809	0.4466052	−1.015	−0.974	0.2079 ***	0.0919 **	0.1390 **	0.1063 *	0.1052 *	0.1505 ***	0.0235	−0.0726	0.0527	0.1481 ***	1.0000		
12. SME	0.8035019	0.3977365	−1.532	0.349	−0.0374	−0.0153	−0.0718	0.0037	0.0074	−0.0259	−0.0091	−0.2010 ***	−0.1421 ***	−0.0143	−0.0736 *	1.0000	
13. Company age	2.651751	1.172012	3.111	7.71	0.0103	−0.0214	−0.0628	−0.0265	−0.0486	−0.0245	0.0541	0.3258 ***	0.2320 ***	0.0030	−0.0116	−0.4105 ***	1.0000

Source: Own elaboration. Significance levels: * *p* < 0.10. ** *p* < 0.05. *** *p* < 0.0. SME: Small and Medium-sized Enterprises.

**Table 2 ijerph-16-03803-t002:** Variables description.

Variables	Description
Feeling of contribution to productivity	1 if the worker feels they contribute to the organization’s productivity, 0 otherwise.
Scale of feeling contribution to organization’s productivity	1 for workers feeling they don’t contribute to organization’s productivity; 2 for workers feeling they contribute to organization’s productivity to some extent, and 3 for workers feeling they totally contribute to organization’s productivity.
Supervisors’ support	1 if the worker feels satisfied with supervisors’ support/treatment, 0 otherwise.
Good work environment	1 if the worker feels satisfied with the work environment, 0 otherwise.
Professional respect	1 if the worker feels respected by the organization both as a professional and individual, 0 otherwise.
Work-life balance	1 if the worker feels the organization is concerned with work-life balance, 0 otherwise.
Skills development	1 if the worker feels the organization supports skills development, 0 otherwise.
Female	1 if female, 0 otherwise.
Age	1 for 20–25 years; 2 for 26–35 years; 3 for 36–45 years; 4 for 46–55 years; and 5 for ≥55 years.
Married	1 for being married, 0 otherwise.
Manager role	1 for occupying a managing role, 0 otherwise.
College education	1 for having college education, 0 otherwise.
SME	1 for being SME, 0 otherwise.
Company age	1 for 1 to 6 years; 2 for 7 to 15 years; 3 for 16 to 29 years; 4 for 30 to 49 years; and 5 for ≥50 years.

Source: Own elaboration.

**Table 3 ijerph-16-03803-t003:** QWL: Subjective and behavioral components influencing employees’ feeling of contribution to productivity.

Variables	Model 1:	Model 2:	
Dependent Variable: Contribution to Productivity	OLS Regression	Multinomial Logit	
		Baseline: Feeling of not contributing to productivity	
Independent variables:	Coef.	Coef. Feeling of contributing to productivity to some extent	Coef. Feeling of totally contributing to productivity
Supervisors’ support	0.1112487 *** (0.0386135)	0.1387051 (0.2829922)	0.0169725 (0.313576)
Good work environment	0.1012274 ** (0.0396864)	−0.1571931 (0.2944245)	−0.3292686 (0.3255704)
Professional respect	0.1194258 *** (0.0417695)	0.2335013 (0.2996408)	0.5612954 * (0.3395112)
Work-life balance	0.0181309 (0.0371606)	−0.4871505 * (0.2743621)	−0.5201555 * (0.3044264)
Skills’ development	0.0525111 (0.0367527)	0.2142189 (0.271979)	0.2460842 (0.3016579)
Female	−0.0188813 (0.0330991)	0.0149441 (0.2438254)	−0.2331886 (0.2705418)
Age	0.0220647 (0.0191218)	0.3310333 ** (0.1469402)	0.3456309 ** (0.1619994)
Married	−0.0007321 (0.0376591)	−0.2280585 (0.2797668)	−0.0901252 (0.309747)
Manager role	−0.0100354 (0.0443451)	0.4593606 (0.3697954)	0.6808159 * (0.3938579)
College education	0.1415679 *** (0.0379515)	0.0578064 (0.2788375)	−0.0239672 (0.3085947)
SME	0.0022576 (0.045563)	0.1645333 (0.336115)	0.0256681 (0.3730899)
Company age	0.0044527 (0.0160382)	0.0342415 (0.1197577)	−0.0841063 (0.1328729)
Obs.	514	514	
LR Chi^2^	14.38	22.06	
Prob. > Chi^2^	0.0000	0.0002	

Source: Own elaboration. Significance levels: * *p* < 0.10. ** *p* < 0.05. *** *p* < 0.0; Standard errors in brackets. LR Chi^2^: Likelihood Ratio (LR) Chi-Square test; Prob. > Chi^2^: The prob > chi2 statistic for the overall model is a test of the joint null hypothesis that all of the regression coefficients (other than the constant term) are zero.

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
