# Peer review of "Quality of Work Life and Organizational Performance: Workers’ Feelings of Contributing, or Not, to the Organization’s Productivity"

_ijerph, 2019, doi:10.3390/ijerph16203803_

Round 1

Reviewer 1 Report

This was an interesting paper.  I like the fact that you are trying to look at QWL through other lenses to provide more clarity in understanding it and its impacts on organizational life. 

I have a few primarily minor concerns:

I am curious to know if you considered some type of objective measure of actual performance as a sort of “check/comparison” concerning the measure you used for the dependent variable, which was a self-reported subjective measure of “feeling”….  I would like to see some discussion of this - either to address the lack of need for this, or explain why you did not include it. In your development of Hypothesis 4, I am not sure I like the wording of Hypothesis 4 – work-life balance is about more than simply getting to take time off. I am curious if I am reading this sentence correctly or not - “ In each organization, a contact person was identified to ensure completion of the questionnaire, which was afterwards validated or not by the research team.” I assume you simply mean that the questionnaires were validated by the team after? You mention that there was no support for H5 concerning support for continuous learning and skills development.  You then go on to say in your conclusions that “instilling a climate of continuous learning for the collaborator” should be a strategic priority. Understandable that your findings are contrary to other studies you reviewed that have evidence to support that recommendation, but your evidence does not support that recommendation and I don't think it should be included.

Author Response

Quality of Work Life and Organizational Performance: Workers’ feelings of contributing, or not, to the organization’s productivity

Manuscript ID: ijerph-609176

Dear Editor-in-Chief of the International Journal of Environmental Research and Public Health

Prof. Dr. Paul B. Tchounwou

Firstly, we would like to acknowledge the editor and all the reviewers’ comments.

Secondly, we are very pleased with the possibility to revise and resubmit the paper. Considering the answers to the questions raised by the reviewers, we give an overview of what was changed according to each referee’s proposals and constructive suggestions.

Reviewer 1 (Rev.1)

Rev.

Comment No.

Page

No.

Section

Comments Reviewer 1

Answers and amendments

Rev.1.1

2

15

1.

5.

I am curious to know if you considered some type of objective measure of actual performance as a sort of “check/comparison” concerning the measure you used for the dependent variable, which was a self-reported subjective measure of “feeling”…. I would like to see some discussion of this - either to address the lack of need for this, or explain why you did not include it.

We acknowledge the reviewer’s comment. Unfortunately, the original questionnaire does not include other objective measures that can be used in empirical applications as dependent variables that represent organizational performance, since the guiding aim of the research was to study the distinct modes of influence of the QWL’s components on workers' feelings of contribution to the organization's productivity, considering a subjective measure for representing the previously referred feelings.

For addressing the need for elaborating the discussion on the lack of objective measures for representing the dependent variable, that is, organizational performance, the following amendments were done:

(i)               In the item: 1. Introduction:

Despite the limitations associated with the use of this dependent variable with subjective nature, its use seems to be justified, on the one hand, given the lack of studies using the behavioral lens to study the relationship between QWL and organizational performance. On the other hand, as it is not the objective of the present study to compare the relationships and the associated significance, using objective measures versus subjective measures, for the purposes of representing the dependent variable: organizational performance.

(ii)             In the item: 5. Conclusions:

In a related vein, this opens an avenue for tracing further research endeavors, expanding both the number of objective and subjective metrics, in order to gauge the hypothetical differences in the relationships established between QWL’s components and organizational performance, “measured” in objective or subjective terms.

Rev.1.2

6

2.2.

In your development of Hypothesis 4, I am not sure I like the wording of Hypothesis 4 – work-life balance is about more than simply getting to take time off.

We acknowledge the reviewer’s comment. The following amendment was done:

Hypothesis 4 (H4): Workers who have the possibility to enjoy the adoption of work-life balance practices in their organizations, are more likely than others to feel that they contribute to the organization’s productivity.

Rev.1.3

7

3.1

I am curious if I am reading this sentence correctly or not - “In each organization, a contact person was identified to ensure completion of the questionnaire, which was afterwards validated or not by the research team.” I assume you simply mean that the questionnaires were validated by the team after?

We acknowledge the reviewer’s comment. The following amendment was done:

In each organization, a contact person was identified to ensure completion of the questionnaire, which was afterwards validated by the research team.

Rev.1.4

15

5.

You mention that there was no support for H5 concerning support for continuous learning and skills development. You then go on to say in your conclusions that “instilling a climate of continuous learning for the collaborator” should be a strategic priority. Understandable that your findings are contrary to other studies you reviewed that have evidence to support that recommendation, but your evidence does not support that recommendation and I don't think it should be included.

We acknowledge the reviewer’s comment. The sentence mentioned by the reviewer was removed from the item: 5. Conclusions; and the following amendment was done:

(3) ensuring that supervisors protect collaborators from hazardous conditions, to reduce feelings of uncertainty and risk; and (4) giving importance to the duties and tasks performed by collaborators.

We look forward to hearing from you.

Yours sincerely,

October 3, 2019

The authors

Reviewer 2 Report

The theoretical background of the topic needs to be expanded. More detailed explanation and theoretical derivation is needed to formulate the hypotheses. H1 and H1A are acceptable, but H1B is not. Employees who feel strong support of their colleagues are more integrated into the organizational culture; they may make stronger group cohesion, and have stronger levels of affiliation motives. However, according to our knowledge, this is completely independent of the organization’s productivity. If the authors keep this statement, then much more thorough argumentation is needed. Knowing the hypotheses, the question arises why the employees' contribution to the organizational performance was the observed subject of the research by the authors. More importantly, what the organization (or the leaders) think about the performances of the workers. In the empirical research, the table of the sample characterization makes no sense in this form, contains too much data, takes up much space, and does not add to the results, since these variables were not grouped in the research. Otherwise, the statistical methods used are appropriate for the interpretation of the results. Apart from the theoretical criticisms above, the formulation of the conclusions is correct.

Author Response

Quality of Work Life and Organizational Performance: Workers’ feelings of contributing, or not, to the organization’s productivity

Manuscript ID: ijerph-609176

Dear Editor-in-Chief of the International Journal of Environmental Research and Public Health

Prof. Dr. Paul B. Tchounwou

Firstly, we would like to acknowledge the editor and all the reviewers’ comments.

Secondly, we are very pleased with the possibility to revise and resubmit the paper. Considering the answers to the questions raised by the reviewers, we give an overview of what was changed according to each referee’s proposals and constructive suggestions.

Reviewer 2 (Rev.2)

Rev. Comment No.

Page

No.

Section

Comments Reviewer 2

Answers and amendments

Rev.

2.1

1

3

1.

2.

The theoretical background of the topic needs to be expanded. More detailed explanation and theoretical derivation is needed to formulate the hypotheses.

We acknowledge the reviewer’s comment. In order to provide additional explanation and theoretical derivation of the research hypotheses, funded both in theoretical and empirical literature, the following amendments were done:

(i) In the item: 1. Introduction: new references were incorporated for justifying the selection and innovativeness of the topic, considering the lack of studies exploring the subjective and behavioral components of the QWL, as well as a subjective measure for assessing the organizational performance. The two following paragraphs were introduced:

The most sensitive components of the QWL, still unexplored, are intrinsically related to the socio-emotional and psychological needs of employees, which require the application of more behavioral lenses, in order to unveil the components that can most influence job satisfaction and motivation, but also productivity [4,5].

In the context of health organizations, the relationship between QWL and productivity was already investigated, suggesting the design of adequate strategies to reinforce the productivity in hospitals [6]. However, little is known about the different ways in which the behavioral and subjective components of the QWL can influence the employee's feeling of contribution to the productivity of the organization that they integrate.

(ii) In the item: 2. Literature Review and Research Hypotheses; considering the reviewer’s comment, two fundamental changes were done. Firstly, it is presented the item: 2.1. Revealing the Relationship Between Organizational Performance and QWL; and afterwards, the item: 2.2. Exploring Subjective and Behavioral Components of QWL; was revised by incorporating additional references with theoretical and empirical support for the development of the following research hypotheses:

(1)   Hypothesis 1:

Cooperative decision making, adequate recognition and supportive supervisors are considered fundamental to QWL [33], with appropriate job performance feedback and favorable relations with supervisors being said to have a direct impact on QWL [34]. Another study [35] goes further and reveals that supervisory behavior is the most important component of QWL, contributing to the variance in the employee’s role efficacy by as much as 21%.

Considering the previous statements in the literature, the following research hypothesis is derived:

Hypothesis 1 (H1): Workers who feel that they are supported and appreciated by their supervisors are more likely to feel that they contribute to the organization’s productivity.

(2)   Hypothesis 2:

The factors relevant to employees’ QWL include the social environment within the organization, the relationship between life on and off the job, the specific tasks they perform and the work environment [37].

Providing safe and healthy working conditions aims to ensure the employee’s good health, thus, taking measures to improve QWL is expected to increase employee’s motivation ultimately leading to the enhancement of performance and productivity [37].

Accordingly, a work environment that is able to fulfill the employee’s personal needs will lead to an excellent QWL [38].

Thus, the following research hypothesis is considered:

Hypothesis 2 (H2): Workers who feel that they are integrated in a good working environment are more likely than others to feel that they contribute to the organization’s productivity.

(3)   Hypothesis 3:

Employees that feel they are treated with respect by people they work with, and employees who feel proud of their job, increase their feeling of belonging to the company, thus feeling that they are an asset to the organization [45]. Studies [46,47] found that feeling respected is a predictor of QWL, together with self-esteem, variety in daily routine, challenging job, autonomy, safety, rewards and good future opportunities; and as already mentioned an improved QWL is expected to lead to a higher productivity [48].

Considering the previous vision, the QWL construct can be completed by incorporating subjective measures related with employee satisfaction, motivation, involvement and commitment with respect to their lives at work [49]. In the same vein, QWL corresponds to the degree to which individuals are able to satisfy their important personal needs while employed by the firm. This gives rise to the following research hypothesis:

Hypothesis 3 (H3): Workers who are respected as professionals are more likely than others to feel that they contribute to the organization’s productivity.

(4)   Hypothesis 4:

Work-life balance has been positioned in the reference literature as a key component of QWL [37,54–58], but it deserves to be noted that the employee’s level of emotional intelligence could influence his/her work-life balance [59].

It should be noted also that in a previous empirical study [60] no significant association, neither positive nor negative, between work-life balance and productivity was detected.

Nevertheless, Work-life balance plays an important role in overall life satisfaction and influences experiences in work life by increasing job satisfaction and organizational commitment [61]. A high level of engagement in work life is likely to produce a positive effect in work-life balance, which can be further enhanced by goal attainment in work life [62]. Accordingly, the following research hypothesis is derived:

Hypothesis 4 (H4): Workers who have the possibility to enjoy the adoption of work-life balance practices in their organizations, are more likely than others to feel that they contribute to the organization’s productivity.

(5)   Hypothesis 5:

Skills, occupational improvement and opportunity for training are considered sub components of QWL [45,64,65]. The development of skills and abilities can improve job satisfaction and overall QWL, and for its turn QWL can influence the employee’s performance [66,67]. Thus, employees expect to develop their skills and get promoted, ensuring a better performance for the organization [68]. In turn, training is an activity aimed at enhancing performance, by ensuring the opportunities for development of skills and encouragement given by the management team [37].

As previously revealed through the empirical evidence obtained in [69], both QWL and motivation influence employees’ performance positively. High levels of QWL lead to job satisfaction, which ultimately results in effective and efficient performance [49]. Considering the previous statements and empirical evidence, the following hypothesis is derived:

Hypothesis 5 (H5): Workers who feel that their organizations invest in their careers, for example through continuous learning, the development of new skills or supporting professional growth, are more likely to feel that they contribute more than others to the organizations’ productivity.

Rev.

2.2

5

2.2

H1 and H1A are acceptable, but H1B is not. Employees who feel strong support of their colleagues are more integrated into the organizational culture; they may make stronger group cohesion, and have stronger levels of affiliation motives. However, according to our knowledge, this is completely independent of the organization’s productivity. If the authors keep this statement, then much more thorough argumentation is needed.

We acknowledge the reviewer’s comment. Thus, we opted to remove the original H1B and consider instead a research hypothesis focused on the role of suppervisors enuntiated as follows:

Hypothesis 1 (H1): Workers who feel that they are supported and appreciated by their supervisors are more likely to feel that they contribute to the organization’s productivity.

This option led us to review the operational model of analysis (please see Figure 1: QWL and Feeling of Contribution to Productivity: Operational model of analysis), as well as to a new estimation of the model specification, which provided similar descriptive statistics and significant results that are presented in Tables 2 and 3, which were also renumbered and revised, accordingly.

Rev.

2.3

15

5.

Knowing the hypotheses, the question arises why the employees' contribution to the organizational performance was the observed subject of the research by the authors. More importantly, what the organization (or the leaders) think about the performances of the workers.

We acknowledge the reviewer’s comment. Thus, we would like to clarify that the original questionnaire focuses solely on the analysis of workers, evaluating their feelings of contribution to the productivity of the organization. It is worthwhile to note that it was defined by the research team that no managers, partners or employers would be interviewed in order to avoid any possible bias concerning the free expression of the workers' feelings.

Moreover, the reviewer’s comment is very appreciated in the sense that it can be used as a future research line, which will imply the design of a new questionnaire targeted to assess the feelings of the leaders regarding the performance of workers, and, afterwards, it will be possible to produce a contrasting analysis.

For addressing this last comment, and additional sentence was incorporated in the item: 5. Conclusions; for raising an additional future research endeavor:

This would imply the design of a new questionnaire targeted to assess the feelings of the leaders regarding the performance of workers, and, afterwards, it will be possible to produce a contrasting analysis.

Rev.

2.4

8

3.2.1.

In the empirical research, the table of the sample characterization makes no sense in this form, contains too much data, takes up much space, and does not add to the results, since these variables were not grouped in the research.

We acknowledge the reviewer’s comment. The table of the sample characterization was removed from the revised version of the manuscript and the subsequent tables were renumbered according to this change. In addition, the item 3.2.1. was retitled as: 3.2.1 Sample and descriptive statistics.

Rev.

2.5

Otherwise, the statistical methods used are appropriate for the interpretation of the results. Apart from the theoretical criticisms above, the formulation of the conclusions is correct.

We are grateful for the reviewer’s comment and positive feedback.

We look forward to hearing from you.

Yours sincerely,

October 3, 2019

The authors

Round 2

Reviewer 2 Report

Thank you for your revision.